# Subcritical Water Extraction for Valorisation of Almond Skin from Almond Industrial Processing

**DOI:** 10.3390/foods12203759

**Published:** 2023-10-13

**Authors:** Pedro A. V. Freitas, Laia Martín-Pérez, Irene Gil-Guillén, Chelo González-Martínez, Amparo Chiralt

**Affiliations:** Institute of Food Engineering FoodUPV, Universitat Poltècnica de València, 46022 Valencia, Spain

**Keywords:** phenolic compounds, cellulose fibres, integral fractionation, active compounds, bleaching optimisation, bioactive properties

## Abstract

Almond skin (AS) is an agro-industrial residue from almond processing that has a high potential for valorisation. In this study, subcritical water extraction (SWE) was applied at two temperatures (160 and 180 °C) to obtain phenolic-rich extracts (water-soluble fraction) and cellulose fibres (insoluble fraction) from AS. The extraction conditions affected the composition and properties of both valorised fractions. The dry extracts obtained at 180 °C were richer in phenolics (161 vs. 101 mg GAE. g^−1^ defatted almond skin (DAS)), with greater antioxidant potential (1.063 vs. 1.490 mg DAS.mg^−1^ DPPH) and showed greater antibacterial effect (lower MIC values) against *L. innocua* (34 vs. 90 mg·mL^−1^) and *E. coli* (48 vs. 90 mg·mL^−1^) than those obtained at 160 °C, despite the lower total solid yield (21 vs. 29%) obtained in the SWE process. The purification of cellulose from the SWE residues, using hydrogen peroxide (H_2_O_2_), revealed that AS is not a good source of cellulose material since the bleached fractions showed low yields (20–21%) and low cellulose purity (40–50%), even after four bleaching cycles (1 h) at pH 12 and 8% H_2_O_2_. Nevertheless, the application of a green, scalable, and toxic solvent-free SWE process was highly useful for obtaining AS bioactive extracts for different food, cosmetic, or pharmaceutical applications.

## 1. Introduction

Almonds are a very important crop throughout the world’s temperate regions, with a worldwide almond production of 4.6 Mt in 2021, with the USA being the first producer with about 2.0 Mt per year (*International Nuts and Dried Fruits Statistical Yearbook*). Fresh almond fruit consists of four portions: kernel or meat (~11%), middle shell (~33%), outer green shell cover or almond hull (~52%), and a thin leathery layer known as brown skin (~4%) [1]. The processing by-products, shells, and hulls, account for more than 50% of the dry weight of the almond fruits, which, in the past, were used as animal feed or burned for fuel production and energy use. Likewise, in the industrial process of almond kernel, the brown skin is removed by blanching, generating a residual product that constitutes about 6–8% wt. of the seed in large amounts, which has been also used as animal feed or burned as fuel in processing plants [2].

However, almond skins (AS) are known to have a great number of nutritional and health benefits, mainly based on the presence of polyphenols and the high dietary fibre content [3]. Different functional properties have been reported for these phenolic compounds, such as inhibiting lipid oxidation by scavenging free radicals, chelating metals, activating antioxidant enzymes, reducing tocopherol radicals, or inhibiting enzymes that cause oxidation reactions [4]. The composition of AS with high phenolic content and different functional properties represents a high added value for this industrial by-product. Several authors [1,4,5] have reviewed the phenolic composition of AS reported in different studies using different extraction methods. The major components of AS are proanthocyanidins (dimers or larger molecules) (mean content: 6.98 mg.100 g^−1^), hydrolysable tannins, flavonoids (non-isoflavone) (71.3 mg.100 g^−1^), and phenolic acids and aldehydes (11.6 mg.100 g^−1^) [5]. In the complex phenolic profile of AS, different flavonol glycosides have been identified, namely, isorhamnetin rutinoside, isorhamnetin glucoside, kaempferol rutinoside, kaempferol glucoside, quercetin glycosylated to glucose, galactose and rhamnose, kaempferol, naringenin, catechin, protocatechuic acid, vanillic acid, and a benzoic acid derivative [4]. Flavanols and flavonol glycosides were the most abundant phenolic compounds in AS, representing up to 38–57% and 14–35% of the total quantified phenolics, respectively [6]. Due to their potential antioxidant properties, AS can be considered to be a value-added by-product for potential use as dietary antioxidant ingredients. Esfahlan et al. (2010) [4] reported antioxidant activities of ethanolic extracts from AS using different free radical scavenging assays revealing that the Trolox equivalent antioxidant capacity of the AS extract was 13 times greater than that of the whole seed extract at the same extract concentration.

AS also has antimicrobial properties that may arise from the action of phenolic compounds or their synergistic interactions. Methanolic extracts of defatted AS exhibited antibacterial activity against *L. monocytogenes* and *S. aureus* in the range of 250–500 μg mL^−1^, with non-blanched skins showing antimicrobial potential against the Gram-negative *S. enterica* [7]. The aglycones epicatechin and catechin isolated from AS showed great activity against *S. aureus* ATCC 6538P (MIC values of 0.078–0.15 and 0.15 mg.mL^−1^, respectively) [8]. Extracts from liquid-nitrogen-blanched skin also inhibited *Escherichia coli, Serratia marcescens*, and *Streptococcus mutans* [9].

In terms of health benefits, phenolic compounds of AS are bioavailable and extensively biotransformed by the microbiota and host tissue upon consumption [10]. Although the precise function of polyphenol metabolites in preventing chronic disease is emerging, these have been described as protective agents against inflammatory processes [3], cancer [11], herpes [8,12], or cardiovascular disease [13,14], while modulating plasma biomarkers of oxidative stress [15]. AS also exhibits prebiotic activity [16] due to the polyphenol delivery to the colon [17,18] and increased probiotic bacteria in the culture, including bifidobacteria and eubacteria [19]. Thus, almond polyphenols may contribute to gut health through modulation of microbiota.

On the basis of the described properties of AS and their availability as industrial waste from almond processing, obtaining active extracts from AS for pharmaceutical cosmetics or food industry applications represents an interesting strategy for its valorisation. In this sense, sustainable, ecologic, and scalable extraction methods should be proposed and developed. At the laboratory scale, solid–liquid extraction and liquid–liquid extraction steps have been applied using organic solvents, such as methanol, acetone, or ethyl acetate to obtain phenolic-rich extracts [20]. Microwave-assisted extraction of AS has been optimised to increase the phenolic yield, using ethanol as the extraction solvent [21]. Phenolic compounds from gamma-irradiated AS were extracted with 40% ethanol, obtaining an increase in the yield of total phenolic content and enhanced antioxidant activity of the extracts [22]. Nevertheless, a chemical-free extraction process would be more interesting from the point of view of its ecology and sustainability. The overuse of organic solvents causes problems, such as the negative environmental impact through the generation of toxic wastes, the solvents’ effect on human health, and the labour-intensiveness and time requirements of the extraction methods [23]. Subcritical water extraction (SWE), also known as pressurised hot-water extraction or superheated water extraction, is a green technique for the isolation of organic compounds and hemicellulose fractions with preserved molecular functionalities and high molecular weight [24]. Under subcritical conditions, the water solvent properties are similar to those of organic solvents, being able to dissolve different components of medium and low polarity [25] due to the changes in the water properties, depending on different factors. Temperature is the most important factor affecting the extraction efficiency and selectivity. When temperature increases, the dielectric constant and the polarity of water decrease, which enhances the solubility of less polar compounds in water. Thus, subcritical water can behave similarly to methanol or ethanol [26]. This makes subcritical water a green extraction fluid used for a variety of organic species. Compared with organic solvents, subcritical water not only has advantages in ecology, economy, and safety, but also its density, ion product, and dielectric constant can be adjusted by temperature [25]. Likewise, it is faster and scalable, does not require pre-treatments, and produces a low degree of sugar degradation [25,26,27]. It has been applied to extract polyphenols from several natural products such as pistachio hulls [28], chestnut shells [29], comfrey root [30], or pomegranate peel [31]. The optimal extraction temperature and time deduced from different studies were between 130 and 200 °C and 15 and 35 min [26].

When applied to extract natural products, the extraction residue could also be used to obtain insoluble polysaccharides, such as cellulose for different applications, enabling the integral fractionation of material. Different studies have been carried out using SWE for the integral fractionation of plant products, such as rice husk, rice straw, or grapevine by-products, in order to achieve a comprehensive use of the starting material [23,32,33,34]. Nevertheless, no previous studies were found on the AS by-product fractionation by SWE to obtain value-added components for specific applications.

The aim of this work was to fractionate blanched AS industrial residue using SWE at two temperatures (160 and 180 °C) to obtain phenolic-rich extracts and polysaccharide-rich solid residues for different uses. Phenolic content, antioxidant, and antibacterial properties were characterised in the extracts, while the potential use of the residue as a source of cellulose fibres has been explored, through cellulose purification by a green bleaching process using hydrogen peroxide. Active extracts and cellulose fibres could be used to obtain active, reinforced packaging materials.

## 2. Materials and Methods

### 2.1. Chemicals

Folin–Ciocalteu reagent (2 N), gallic acid, methanol, 2,2-Diphenyl-1-picrylhydrazyl (DPPH), sodium hydroxide (NaOH), glucose, and arabinose were purchased from Sigma-Aldrich (USA). Ethanol (98%), hydrogen peroxide (H_2_O_2_, 30%), sulphuric acid (H_2_SO_4_, 98%), and sodium carbonate (Na_2_CO_3_, 99.5%) were obtained from Panreac Quimica S.L.U (Castellar del Vallés, Barcelona, Spain). Phosphorous pentoxide (P_2_O_5_, 98.2%) was obtained from VWR Chemicals (Leuven, Belgium). D(+)-Xylose was supplied by Merck KGaA (Darmstadt, Germany). For microbiological analyses, tryptone soy broth (TSB), bacteriological agar, and peptone water were purchased from Scharlab (Barcelona, Spain). Strains of *Escherichia coli* (CECT 101) and *Listeria innocua* (CECT910) were obtained from the Spanish Type Collection (CECT, Universitat de València, València, Spain).

### 2.2. Plant Material Preparation

Blanched ASs (*Prunus dulcis*, *Nonpareil var.* from California) were kindly provided by Importaco S.A (Valencia, Spain) from its almond peeled production line as an agro-industrial by-product from the harvest of 2022. The plant material was dried at 40 ± 2 °C for 3 days in a forced-air oven (S. P. Selecta, s. a., Barcelona, Spain). Afterwards, the dried AS was milled using a mill (Model SM300 stainless, Retsch GmbH, Haan, Germany) and sieved to obtain particles with dimensions of under 0.55 mm and stored until further use.

Given the high oil content, the dried AS was defatted using petroleum ether (40–60 °C bp) with Soxhlet. The residual solvent in the defatted almond skins (DAS) was eliminated at 40 °C for 3 days in a fume extraction cabin. Meanwhile, the almond oil (AO) (soluble fraction) was recovered using a vacuum rotary evaporator (Rotary Evaporators, Heidolph Instruments GmbH & Co. KG, Walpersdorfer, Germany) at 35 °C, and the total oil content of AS was quantified.

### 2.3. Obtaining AS Fractions by Applying SWE

DAS was subjected to subcritical water extraction (SWE) using a pressure reactor (Model 1-TAP-CE, 5 L capacity, Amar Equipment PVT. LTD, Mumbai, India) at a DAS: distilled water ratio of 1:6 (*w*/*w*). Two extraction conditions were applied according to previous studies into phenolic extraction of plant products [15,34]: 160 °C, 7 bar, 150 rpm, for 30 min and 180 °C, 15 bar, 150 rpm, for 30 min. After each extraction procedure, the DAS dispersions were filtered (Filterlab, Barcelona, Spain, pore size < 0.5 mm) and the solid extracts were obtained by freeze-drying the soluble fractions at −60 °C, 0.8 mbar for 72 h. The obtained extracts, named E-160 and E-180, were stored in desiccators (P_2_O_5_, 0% relative humidity (RH)) at 4 °C until further use. In every SWE condition, the total solid yield of the extracts (TSY, g extract solids/100 g DAS) was determined by sampling two aliquots of the liquid extracts and drying at 105 °C until constant weight to determine the solid:water ratio and the corresponding total solids extracted from DAS, considering the total water mass in the reactor. The insoluble residues (R-160 and R-180) were washed with distilled water, filtered, and dried at 40 °C for 48 h, to determine the yield in the extraction residue, and afterwards stored at 5 °C until further use.

### 2.4. Bleaching Process of the Extraction Residue

The extraction residues (R-160 and R180) were submitted to a bleaching process in order to obtain cellulose fibres, following the methodology described by Li et al. (2011) [35], with some modifications. In order to optimise the bleaching process, hydrogen peroxide solutions at 4% and 8% were used in a bleaching solution, with a solid residue ratio of 30:1. The temperature of the dispersions was kept at 40 °C (using a water bath), while the tested reaction times were 1, 2, and 3 h, fitting the solution pH values at 11 (0.75% wt. NaOH) and 12 (2.15% wt. NaOH). After the bleaching time, the cellulose fractions were filtered and washed with abundant distilled water to remove residual bleaching solution and dried at room temperature for 2 days. To determine the bleaching efficiency of each condition applied, the bleached samples were characterised in terms of yield (Equation (1)) and whiteness index (*WI*). For this, the colour coordinates *L** (lightness), *a** (red-green), and *b** (yellowish-blue) of each bleached fraction were obtained with a CM-3600d spectro-colourimeter (Minolta Co., Tokyo, Japan), and the *WI* was calculated according to Equation (2).
(1)$$Yield=\frac{Weight\;of\;bleached\;cellulose}{Weight\;of\;material\;before\;bleaching}\;\;$$
(2)$$WI=100-\sqrt{{\left(100-L^{*}\right)}^{2}+a^{*2}+b^{*2}}\;$$

Once the optimum bleaching conditions (lowest yield and highest *WI*) were selected, four bleaching cycles were applied to the solid fractions to enhance the cellulose fibre purification process.

### 2.5. Characterisation of Cellulosic Residues

#### 2.5.1. Chemical Composition

The chemical composition of the raw material (DAS), extraction residues (R160 and R180), and bleached fractions were analysed in terms of structural carbohydrates, lignin, and ashes. Before the quantification, the extractives were determined using the standard NREL method (NREL/TP-510-42619—2008) [36]. This procedure was performed using a Soxhlet set-up, which consists of two stages: a first extraction with water for 6 h, followed by a second extraction with ethanol at 60 °C for 6 h. Once the material was free of extractives, structural carbohydrates, and lignin were determined following the procedure described in the standard NREL method (NREL/TP-510-42618—2008) [37], which consists of a two-step acid hydrolysis with H_2_SO_4_. The insoluble fraction was used to gravimetrically determine the Klason lignin content. In addition, the soluble fraction was used to quantify the sugar content (glucose, xylose, and arabinose) using high-performance liquid chromatography (HPLC, Agilent Technologies, model 1120 Compact LC, Waldbronn, Germany). A RezexTM RCM-Monosaccharide Ca^2+^ column (150 × 7.8 mm) and an evaporative light scattering detector (ELSD Agilent Technologies 1200 Series, Waldbronn, Germany) were used. The mobile phase was deionised water, in an isocratic mode, at a flow rate of 0.4 mL.min^−1^. The detector conditions were: 40 °C, 3.0 bar gas pressure (N_2_), and a gain of 3. The software used was ChemStation (version LTS 01.11, Agilent Technologies, Waldbronn, Germany) and the data were analysed with the Origin program (version OriginPro 2021, OriginLab Corporation, Northampton, MA, USA) by applying the Gaussian model for peak area determination. The cellulose content was determined from the glucose content, while the hemicellulose content was obtained from the sum of xylose and arabinose sugars.

The ash content of the cellulosic fractions was determined by sample incineration at 575 °C for 24 h. The procedure was performed in duplicate.

#### 2.5.2. Thermogravimetric Analysis (TGA)

The thermal stability of different samples was analysed using a thermogravimetric analyser (TGA 1 Stare System analyser, Mettler-Toledo, Greifensee, Switzerland). In each case, 4–5 mg of conditioned samples (in desiccators containing P_2_O_5_, at 25 °C for two weeks) were weighted in alumina pans and heated from 25 °C to 900 °C under a constant flow of nitrogen (10 mL.min^−1^) and a 10 °C.min^−1^. Samples were analysed in duplicate and TGA and DTGA curves were obtained using the STAR^e^ Evaluation Software (version V12.00a, Mettler-Toledo, Inc., Greifensee, Switzerland).

### 2.6. Characterisation of SWE Extracts

Phenolic-rich extracts obtained by SWE (E-160 and E-180) were characterised according to total phenolic content, antioxidant activity, and antibacterial activity.

#### 2.6.1. Determination of Total Phenolic Content (TPC)

Total phenolic content (TPC) of the solid extracts was determined using the Folin–Ciocalteu method, according to the procedure described by Freitas et al. (2020) [38]. Briefly, 0.5 mL of each extract was mixed with 6 mL of distilled water and 0.5 mL of Folin reagent (2 N). After one minute, 1.5 mL of Na_2_CO_3_ at 20% (*w*/*v*) was added and the final volume was adjusted to 10 mL with distilled water and kept in the dark at room temperature for 2 h. Afterwards, the absorbance was measured using a UV-Vis spectrophotometer (model Evolution 201, Thermo Scientific, Waltham, MA, USA) at 747 nm. The total phenolic content was determined using a standard curve (*R*^2^ = 0.9991) of gallic acid (2–20 mg·L^−1^) and expressed as mg gallic acid equivalents (GAE).100 g^−1^ extract. The measurements were performed in triplicate.

#### 2.6.2. Radical Scavenging Activity (DPPH)

The anti-radical activity of the extracts was determined using the free radical 2,2-diphenyl-1-pikryl-hydrazyl (DPPH) method according to a modified procedure described by Freitas et al. (2020) [38]. For each extract, methanolic DPPH solution at 6.22 × 10^−2^ mM (Abs at 515 nm = 0.7 ± 0.2) was mixed with different extract concentrations to a final volume of 4 mL. The resultant solutions were kept in the dark at room temperature for 12 h, and then the absorbance at 515 nm was measured. The initial and final concentration of DPPH in the reaction medium was calculated from a calibration curve fitted by linear regression (*R*^2^ = 0.9992). The anti-radical activity was evaluated by means of EC_50_, defined as the amount of antioxidants required to reduce the initial concentration of DPPH by 50%, when reaction stability is reached, and expressed as mg freeze-dried.mg^−1^ DPPH. The EC_50_ values were determined from graphics of %[*DPPH*]_*remaining*_ vs. mg solid extract/mg DPPH:(3)$$\%{\left[DPPH\;\right]}_{remaining}\;=\frac{{\left[DPPH\right]}_{t}}{{\left[DPPH\right]}_{t=0}}\;\times \;100\;\;\;\;\;$$
where [*DPPH*]*_t_* is the DDPH concentration value when the reaction was stable, and [*DPPH*]*_t_*_=0_ is the DPPH initial concentration. Each determination was carried out in duplicate.

#### 2.6.3. Antibacterial Activity

The minimum inhibitory concentration (MIC) of the extracts (E-160 and E-180) was determined for two bacteria: a Gram-positive bacterium *Listeria innocua* and a Gram-negative bacterium *Escherichia coli*. This analysis was carried out following the method described by Requena et al. (2019) [32] and Freitas et al. (2020) [38]. For this analysis, standard 96-well microtiter plates with a total volume of 200 μL were used.

For both bacterial strains, previously stored at −20 °C, a stock solution was prepared by transferring amounts of bacteria using an inoculation loop to a volume of 10 mL TSB and incubated at 37 °C for 24 h. A volume of 10 μL of the stock solution was taken and transferred to a tube containing 10 mL of TSB to prepare the corresponding working solution with 10^5^ CFU.mL^−1^, which was confirmed by serial dilution and counting. For each bacterium, 100 μL of the bacterial solution with an initial concentration of 10^5^ CFU.mL^−1^ was added to each well. Different volumes of each extract solution at 200 mg.mL^−1^ were added, and then the final volume of each well was made up to 200 μL with TSB, and the plates were incubated at 37 °C for 24 h. Afterwards, 100 μL of each well was transferred in TSA plates and incubated at 37 °C for 24 h for final counting. For each extract, the MIC was determined as the lowest extract concentration at which no bacterial growth was observed on the TSA plate. This analysis was performed in duplicate.

#### 2.6.4. Protein Content

The protein content was determined by the Dumas method using a Leco instrument calibrated according to the manufacturers. The analysis was performed in duplicate for DAS, extracts (E-160, E-180), and residues (R-160 and R-180) obtained from SWE. The factor to convert total nitrogen in protein content was 6.25.

### 2.7. Statistical Analysis

The experimental data were subjected to an analysis of variance (ANOVA) using Minitab statistical software (version 17), considering a confidence level of 95%. Differences between treatment responses were determined by the Fisher test, using the least significant difference (α) of 5%.

## 3. Results and Discussion

### 3.1. Yields of the Process Steps

Figure 1 shows the flow chart of the process applied to AS fractionation indicating the obtained yield of each process step. In the first defatting step, 20.8 ± 0.7% of almond oil was obtained from the dried AS, which represents an important amount of oil with high value in the food and cosmetic market. This content was similar to that reported in other studies using AS [39]. The lipid composition of blanched AS has been reported [39], where monounsaturated fatty acids (mainly oleic acid) are the principal component (56%), followed by polyunsaturated (34%, mainly linoleic acid) and saturated fatty acids (10.3%, mainly palmitic acid). The reported total vitamin E was 13 mg.100 g^−1^, of which α-tocopherol represented 99%; only traces of β- and γ-tocopherol were detected. These data were consistent for samples from different sources, almond cultivars, and varieties [39]. The lipid profile of AS is similar to that observed in almond kernel oil and could be extracted for different applications [40].

The yields in terms of solid extracts (E-160 and E-180) and dried residues (R-160 and R-180) of the SWE process performed at 160 and 180 °C are shown in Figure 1. In contrast with that obtained for other plant materials, where extraction yield increased when temperature rose [28,34,41], the extract yield from DAS was lower at 180 °C (21 g dry extract.100 g^−1^ DAS) than at 160 °C (29 g dry extract.100 g^−1^ DAS). Likewise, this trend was also observed for the yield in residues after extraction (56 g dried insoluble fraction.100 g^−1^ DAS and 55 g dried insoluble fraction.100 g^−1^ DAS). It is also remarkable that the sum of both yields at a given temperature does not close the mass balance, in which about 15% and 24% of dry matter was not recovered at 160 °C and 180 °C, respectively. On the other hand, the pressure in the extraction reactor exceeded the water vapour pressure at the set point temperature by 2 and 5 bars, for 160 and 180 °C, respectively. This result suggests the formation of gases associated with a certain degree of mineralisation of the organic matter present in the reactor at the process conditions, forming CO_2_. In fact, SWE has been applied with high efficiency in the mineralisation of contaminant organic compounds, using hydrogen peroxide at different concentrations to effectively remove many hazardous compounds in wastewater [42,43]. Although no oxidising agent was added in the extraction carried out, and it took place at lower temperatures, the demonstrated prooxidant action of many phenolic compounds [44], depending on the medium conditions, could induce partial mineralisation of DAS samples under the subcritical water conditions applied in the study; the higher the extraction temperature, the greater the substrate degradation.

Other authors also observed thermo-degradation of different compounds when temperature increased in the SWE. For instance, procyanidin dimers were identified in pistachio extracts, indicating the thermal degradation of compounds of larger molecular weights [28]. The application of high temperature (190 °C) and long extraction time led to the modification or degradation of stilbenes from grapevine by-products [23]. Likewise, 5-hydroxymethylfurfural (HMF) formation was observed above 150 °C due to thermo-oxidation or Maillard reactions occurring at high temperatures [45,46]. The formation of HMF at high temperatures might be attributed to the decomposition of monomeric sugars formed through the hydrolyses of polysaccharides, such as hemicelluloses, which is promoted by SWE at high temperatures [27,32,47].

The thermogravimetric analysis of the samples (DAS, extracts, and residues), shown in Figure 2, was coherent with the partition of components between the liquid and solid phases during extraction. The raw DAS presents three thermal degradation steps: the first one between 40 and 180 °C corresponds to the loss of adsorbed water and reaction products due to the caramelisation of free sugars present in the plant matrix [33]; the second one (190–450 °C) is associated to the degradation of polymeric materials, which are constitutional polysaccharides such as hemicellulose (150–350 °C) and cellulose (275–350 °C), and a fraction of the total lignin (160–900 °C) [48,49]; and the third one (460–670 °C) relates to the degradation of the residual lignin fraction and the products from the fragmentation of the organic structures thermo-degraded at lower temperatures [50]. Malayil et al. (2022) [51] observed similar thermal degradation behaviour for the untreated AS. The extracts (Figure 2a) were enriched in low molecular weight compounds with lower degradation temperatures, while the insoluble residues (Figure 2b) showed an increase in polymeric material, as deduced from their degradation at higher temperatures. In both cases, an increase in mass loss was observed for the temperature range of lignin degradation, which is consistent with the formation of degraded/condensed compounds during the high-temperature extraction process. An increment in the final mass residue assignable to ash is also observed for both extracts and insoluble residues. However, under nitrogen flow, this final residue could, in part, correspond to degraded/condensed organic matter more temperature-resistant to oxidative processes. Therefore, thermal analysis was coherent with the selective fractionation of DAS by applying SWE, giving rise to aqueous extracts richer in different compounds of lower molecular (sugars, phenolic compounds, and minerals) and polysaccharide-rich insoluble residues.

The ash content of the DAS, extracts, and residues (Table 1 and Table 2), determined by sample incineration, indicated that minerals from DAS were mainly present in the extracts (13.1 and 15.5%, respectively, for E-160 and E-180) whereas small amounts remained in the residues (2.3 and 2.8%, respectively, for R-160 and R-180). Considering these values and the respective yields in extracts and residues, the predicted ash content was similar to that obtained for the DAS sample (5.09%). This value was in the range of the ash content reported by other authors for almond skin [39], which is rich in manganese, calcium, zinc, iron, magnesium, copper, phosphorus, and potassium [52,53].

Subcritical extraction also promoted partition of the protein content of DAS (13.2 ± 0.1 g.100 g^−1^ DAS), which was higher in the extracts (13.6 ± 0.4 and 24.1 ± 0.3 g.100 g^−1^ dried extract, for E-160 and E-180, respectively) whereas the extraction residues became poorer in protein at 180 °C (13.4 ± 0.3 and 12.7 ± 0.4 g.100 g^−1^ dried residue, respectively, for R-160 and R-180). The protein content of DAS was in the range of previously reported values for almond skin [39]. The enrichment of the extraction residues in protein indicates the low solubility of almond skin protein under the used water subcritical conditions.

### 3.2. Antioxidant and Antibacterial Properties of the SWE Extracts

The obtained extracts at 160 and 180 °C were characterised according to their antioxidant and antibacterial properties in order to know their potential applications as functional ingredients in the food, pharmaceutical, or cosmetic industry. Table 1 summarises the total phenolic content (TPC) determined by the Folin–Ciocalteu method and the DPPH radical scavenging capacity (by means of EC_50_), as well as the minimal inhibitory concentration (MIC) for *L. innocua* and *E. coli*.

The TPC determined in the solid extracts (TPC_1_) was also referred to per mass unit of DAS (TPC_2_). Higher values were obtained at 180 °C than at 160 °C, which is in line with that observed by other authors for the phenolic extraction of different plant residues submitted to SWE [25,28,34]. Ersan et al. (2018) [28] observed a very efficient selective extraction of different phenols in pistachio hulls by SWE at 150–170 °C; total gallotannin yielded up to 33 mg.g^−1^, where gallic acid (22.2 mg.g^−1^) and penta-*O*-galloyl-*β*-d-glucose (9.77 mg.g^−1^) levels were 13- and 11-fold higher than those in the aqueous methanol extracts. However, neo-formed antioxidant compounds during extraction at the highest temperature [45,46] could also contribute to the quantified TPC by the unspecific Folin–Ciocalteu method. Vladic et al. (2020) [30] reported high levels of HMF when SWE was applied to pomegranate peel at temperatures above 130 °C for 20 min. The authors recommended lower temperatures of extraction because they provide a high content of phenols and minimise the presence of HMF. Considering the reported maximum non-toxic HMF intake levels (80–100 mg.kg^−1^ per day) [54], the formed levels of this compound should be considered carefully when the extracts are intended to be used as a food additive.

In terms of the TPC extracted from DAS, 2.96 and 3.38 g GAE.100 g^−1^ DAS were obtained at 160 and 180 °C, respectively. These values were higher than those reported by other authors for blanched DAS using different extraction methods based on organic solvents. Mandalari et al. (2010) [39] reported 0.279 g GAE.100 g^−1^ blanched skins, this value is lower than that obtained for non-blanched skins (3.474 g GAE.100 g^−1^), which is mainly due to the removal of water-soluble flavonoids and phenolic acids during the industrial blanching and is similar to that reported by Garrido et al. (2008) [55]. Other authors [56] also reported significant losses (89%) in the TPC of peanut skins during the blanching process. Phenolic compounds from AS also showed oxidative and other degradative phenomena during drying thermal treatment. The total amounts of phenolic compounds extracted from thermally dried skins were always lower than those extracted from non-dried skins, with the most evident decrease affecting (+)-catechin. Therefore, antioxidant capacity may be affected by the degradation of bioactive compounds at high temperatures, generating other products with antioxidant properties, as observed for dried blanched skins and roasted samples [55]. It is worth mentioning that the Folin–Ciocalteu method is based on a redox reaction, where compounds other than phenolics, such as ascorbic acid, reducing sugars, or some amino acids, can reduce the Folin–Ciocalteu reagent, thus overestimating the TPC content [57,58]. Nonetheless, this method is useful for indicating the redox power of extracts obtained under different extraction conditions, which may be related to their global antioxidant capacity. In addition, despite the possible partial modification of the phenolic fraction during SWE, especially at the highest temperature, the process allows for obtaining highly rich extracts in antioxidant compounds, which is probably promoted by removing bonded phenolics in the skin cell structure under subcritical water hydrolysis.

In agreement with the TPC values, the DPPH radical scavenging capacity was also very high. It increased when the extraction temperature rose, exhibiting values near the most known antioxidants, such as ascorbic acid or α-tocopherol (EC_50_: 0.12 and 0.26 mg compound.mg^−1^ DPPH, respectively) [59]. Therefore, the extracts obtained under subcritical conditions, especially the E-180, exhibited a remarkable DPPH radical scavenging capacity, which was close to that of the strong antioxidants. Nevertheless, as previously commented for the TPC values, the increased antioxidant capacities at higher temperatures could be attributed not only to the higher extraction efficiency for antioxidant compounds but also to the neo-formed antioxidants via Maillard, caramelisation, and thermo-oxidation reactions [45,46]. Other antioxidant tests, such as the ABTS radical scavenging capacity or FRAP assay, could be carried out to verify the actual antioxidant potential of the extracts. Nevertheless, for their use as active components of food packaging materials, their effectiveness at controlling food oxidation reactions, such as oxidation of unsaturated lipids, should be proven when incorporated into these materials and brought into contact with the target food.

DAS extracts (E-160 and E-180) also exhibited antibacterial effects against Gram-positive and Gram-negative bacteria. *E. coli* is a Gram-negative bacterium found in the intestines of humans and animals as well as in untreated water and foods, which cause serious food poisoning. Likewise, *Listeria* innocua is a Gram-positive bacterial strain that has previously been used to model the food-borne pathogen *L. monocytogenes* in inactivation experiments [60]. Table 1 shows the MIC values of both extracts for *L. innocua* (90 and 34 mg.mL^−1^ for E-160 and E-180, respectively) and *E. coli* (90 and 48 mg.mL^−1^ for E-160 and E-180, respectively). E-180, with higher TPC values, was more effective against both bacteria since it had lower MIC values, whereas both bacteria exhibited similar sensitivity for each extract. Other authors also reported the antibacterial activity of AS extracts obtained by different procedures, which have been attributed to the action of the constitutive phenolic compounds. Phenolic fractions of almond skin exhibited antibacterial activity against *L. monocytogenes* and *S. aureus* in the range of 250–500 μg.mL^−1^ [7]. The aglycones epicatechin and catechin isolated from AS showed great activity against *S. aureus* ATCC 6538P (MIC values of 0.078–0.15 and 0.15 mg·mL^−1^, respectively) [61]. Smeriglio et al. (2016) [9] report very low MIC values of phenolic extracts from AS against several Gram-positive and Gram-negative strains. Of the Gram-positive bacteria, *S. aureus* showed the highest sensitivity, followed by *S. epidermidis* ATCC 35984 and *S. mutans*. The obtained SWE extracts from DAS could be considered to be effective antibacterial products useful for controlling bacterial growth in different systems, with MIC values lower than that observed for other SWE plant extracts, such as rice straw [34] or rice husk [32].

Therefore, DAS extracts from the almond industrial process, obtained by SWE, can be considered functional ingredients to control oxidative or microbial spoilage of different products, such as foods or pharmaceuticals. Nevertheless, the formation of potentially toxic compounds makes the toxicological study necessary. Likewise, different health-promoting properties of the extracts should be analysed to obtain useful information about their potential uses.

### 3.3. Bleaching of the Extraction Residues

The extraction residues (R-160 and R180) were treated in order to recover the cellulose fraction, which can be used to obtain fibres or nanomaterials useful for different applications [62]. Thus, a green bleaching process using hydrogen peroxide was optimised to purify cellulose, varying the pH, concentration of the oxidant agent, and treatment time. The efficiency of the different bleaching treatments was evaluated by monitoring the yield (BY: mass of bleached residue/initial mass, %) and the whiteness index (*WI*) of the cellulosic material. The yield informs about the total amount of compounds removed in the oxidative process, while the whiteness index accounts for the removal of coloured compounds.

The first experimental bleaching series was carried out with 4% hydrogen peroxide at different pH values (11 and 12) to analyse its effect on the efficiency of the process for 1, 2, or 3 h of treatment. The *WI* and BY values obtained after each procedure are shown in Figure 3a. A multifactorial analysis of variance (MANOVA), with the factors pH, extraction temperature, and bleaching time, revealed a significant effect of pH on BY (F-value: 19) and a significant effect of pH (F-value: 19) and SWE temperature (F-value: 25) on *WI*, with no significant effect of bleaching time in any case. The higher the pH value, the lower the yield (higher removal of oxidisable compounds) and the higher the *WI* (higher removal of coloured compounds). The WI was lower for samples obtained at 180 °C that also showed a lower initial *WI* value.

Based on these results, treatments at pH 12, with 2, 4, and 8% hydrogen peroxide, for 1, 2, and 3 h were carried out. The BY and *WI* values obtained for this experimental series are shown in Figure 3b. The MANOVA showed significantly lower values of BY when the concentration of hydrogen peroxide (F-value: 24) and bleaching time (F-value: 10) rose, while the BY values were significantly lower for the R-180 sample, compared with R-160. Likewise, *WI* showed a significant increase when the oxidant concentration rose, with no significant effect of the bleaching time, exhibiting significantly higher values for the residue obtained at 160 °C. Therefore, the residue obtained at 180 °C was more resistant to bleaching than that obtained at 160 °C, and, in both cases, greater efficiency (lower BY and higher *WI*) was observed with the higher concentration of hydrogen peroxide, but without significant effects of the bleaching time. On the basis of these results, four bleaching cycles were applied at the most effective bleaching conditions (8% hydrogen peroxide, pH 12 for 1 h) to promote cellulose purification, evaluating the BY and *WI* (Figure 4), as well as the composition of the residue in each cycle, in terms of cellulose, hemicellulose, lignin, and ash contents (Table 2). The contents in these components of the raw material (DAS) and the extraction residues (R-160 and R180) were also quantified and are also shown in Table 2.

The application of four successive bleaching cycles significantly decreased BY while *WI* increased, reflecting the progressive purification of cellulose in each cycle (Figure 4). The cellulose purification progress was monitored through the analysis of lignin and sugars through the NREL method. After removal of the water (which includes soluble sugars) and ethanol extractives in the DAS, R-160, and R-180 samples (23.5, 18.5, and 17.3% wt., in total, respectively), glucose, xylose, and arabinose were quantified in the different materials. Glucose was the major component, and arabinose was only present in the DAS sample. Mandalari et al. (2010) [39] reported that carbohydrates comprised 45% of almond skin in which the amounts of different monomeric sugars (as a percentage of total sugar content) were galacturonic acid (36%), glucose (29%), arabinose (18%), and xylose (7.6%). Considering the defatted basis in the DAS sample and the possible free sugars elimination in the previous water extraction, similar contents of glucose (7.7 g.100 g^−1^ DAS), xylose (4.1 g.100 g^−1^ DAS), and arabinose (5.0 g.100 g^−1^ DAS) were obtained in the present study. In the extraction residues (R-160 and R-180), no arabinose was detected, which indicates that it was dissolved under the subcritical water conditions. Hemicellulose content was considered as total xylose content (or xylose and arabinose in the DAS sample), whereas total glucose was attributed to the cellulose content [37].

The percentage of cellulose increased in both extraction residues (R-160 and R-180) (Table 2) with respect to the untreated DAS, and significant differences in cellulose richness were observed for R-160 and R-180, with the former being poorer in cellulose (Table 2). Hemicellulose was more extensively removed from the DAS matrix treated at 180 °C, but lignin and ash contents increased to a greater extent in this case (Table 2). Cellulose content significantly increased in the first and second bleaching cycles for both R-160 and R-180 samples, but no subsequent increase was observed in the successive cycles. This suggests that a part of the quantified glucose was not cellulosic but hemicellulosic and was also progressively removed during blanching.

The Klason lignin content reported for blanched AS was 20.6% wt., but it may be overestimated as the residue representing the Klason lignin is a mixture of lignin, residual protein, suberin, and ash [39]. The quantified lignin content of the DAS sample was in the range of reported values for AS. The high content of Klason lignin determined in both residues (R-160 and R-180) is remarkable, the magnitude of which is unexpected from a mass balance, considering the content of the initial DAS sample. This indicates that the SWE process generated acid-insoluble compounds that were quantified as lignin fractions. Thus, during the partial degradation reactions that occurred in SWE, neo-formed compounds seem to have similar acid-solubilisation behaviour to the initial lignin content, significantly increasing the gravimetrically quantified lignin. This content decreased in the bleached samples, but these still showed very high values, even after the four bleaching cycles, especially in the BR-180 samples obtained a higher extraction temperature, where greater matter degradation was observed in the SWE treatment. The lignin content progressively decreased with bleaching cycles, but the highest initial content of R-180 samples makes its effective removal with four bleaching cycles more difficult.

The enrichment of the samples in the hemicellulose fraction during the first and second bleaching cycles, with no subsequent decrease in the following cycles, is also remarkable. This indicates an inherent difficulty of the material for cellulose purification and removal of other polysaccharides in the lignocellulosic matrix. The maximum percentage of cellulose obtained in the bleached residues was about 50% for BR-180 and 40% for BR-160 by applying only two cycles with no significant subsequent purification in the successive cycles. In fact, only the lignin content progressively decreased throughout the bleaching cycles.

The TGA and their DTGA curves of the insoluble and bleached residues are shown in Figure 5. Compared to the non-bleached residues, the TGA curves of the bleached fractions showed significant differences in the thermal stability associated with the compositional changes that occurred in the bleaching step. The increment of degradable compounds at low temperatures (between 45 and 160 °C) after bleaching, which may correspond to mono or oligosaccharides resulting from the alkaline hydrolysis of the present polysaccharides that are not removed in the subsequent washing step, is remarkable. The degradation of bleached polymeric material is also reflected in the decrease of the degradation temperature of structural polysaccharides. The bleached residues showed a displacement of the main degradation peak from 330 (R-160) and 323 °C (R-180) to a range of 244–267 °C, suggesting the depolymerisation of constitutive polysaccharides during the bleaching process. In fact, the decomposition of the active species (HOO^−^) during the bleaching reactions into its intermediates such as hydroxyl radical (HO•) and superoxide anion (O_2_^−^•) can lead to the attack of the cellulose and hemicellulose chains, thus worsening the thermal stability of the bleached fractions [35,63]. Thermal analysis also showed, the formation of compounds with high thermal stability (600–800 °C) in the bleached residues while decreasing the typical peak of lignin present in the initial residue (460–670 °C). This was observed in all bleached fractions, where the intensity of the characteristic lignin peak decreased as the bleaching cycles progressed, whereas other degradation peaks at higher temperatures (about 725 °C) appeared due to the newly formed condensed compounds. The bleached residues also had a high final residual mass, which increased as the bleaching cycles progressed. This behaviour was similar for both R-160 and R-180 samples when submitted to bleaching. Therefore, the oxidative process applied with hydrogen peroxide in an alkaline medium seems to degrade some constituents that remain bonded to the matrix and do not leach during the bleaching process or subsequent washing, making cellulose purification difficult.

The ash contents of the residues, determined by sample incineration, progressively increased after the successive bleaching cycles, as also observed for the TGA residual mass, reaching values of 21–24% after four bleaching cycles. This behaviour suggests that sodium ions of an alkaline bleaching medium were entrapped in bleached residues, increasing the final ash content of the samples. Other authors [64,65] also observed that when some lignocellulosic matrices were submitted to alkali treatments with NaOH solutions, to promote chemical degradation and removal of amorphous components, Na ions penetrate into the cellulose crystalline network, forming antiparallel crystalline soda–cellulose complexes with modified crystallisation patterns. This allows for the swelling of interfibrillar regions and the rearrangement of the cellulose microfibrils.

Therefore, AS was not a good source of cellulose due to its relatively low content and the difficulties in its purification, in comparison with other agro-industrial residues such as rice straw [33]. In fact, considering the yields of the different steps applied to obtain cellulose (Figure 1), only 11–12% wt. of DAS could be transformed in partially purified cellulose (40–50% purity) by applying SWE and bleaching processes. The highest temperature of SWE allows for the highest purity, but the very low total yield makes the process unprofitable.

## 4. Conclusions

The current increasing trend of almond production and its industrial processing has been accompanied by an increment in bleached skin waste generation, which requires seeking solutions to its reutilisation. The high content of polyphenols in this waste and their bioactivity offer an interesting opportunity for obtaining phenol-rich extracts, using a green and scalable process without using toxic solvents. Subcritical water extraction at 160 or 180 °C allowed for obtaining bioactive extracts (21–29% of defatted almond skin) with high phenolic content and antioxidant and antibacterial properties, which could be used for food, pharmaceutical, or cosmetic applications. The highest temperature provided the extracts with higher bioactive properties. Nevertheless, the thermo-formation of new antioxidant compounds during the partial degradation of extracts could be responsible for this enhanced activity. Therefore, a toxicological study of the extracts is required to ensure their safety for different applications, such as the development of active packaging materials.

Cellulose recovery from the extraction residues (55–56% of defatted almond skin) was not effective enough due to the low yield obtained in the blanching process (20–21%) and the low purity of cellulose (40–50%) reached in the blanched fractions after four bleaching cycles (1 h) at pH 12 and 8% hydrogen peroxide. Therefore, almond skin is not a good source of cellulosic materials.

## Figures and Tables

**Figure 1 foods-12-03759-f001:**
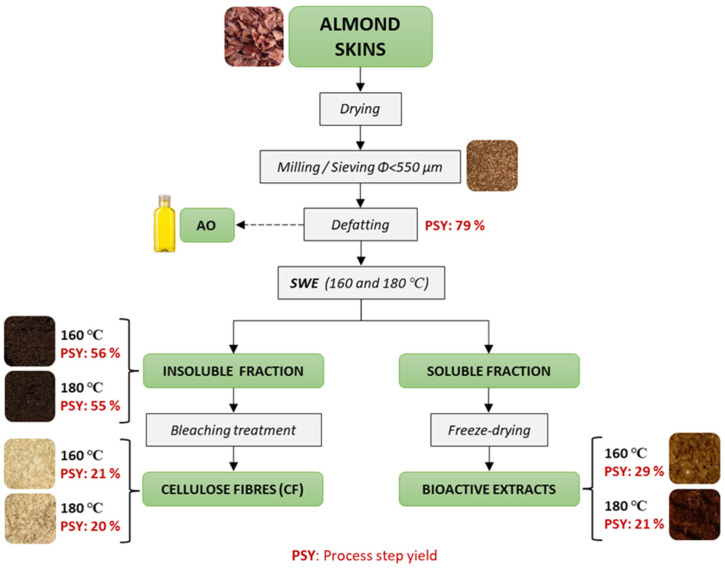
Flow chart of the process used for almond skin (AS) fractionation, showing the process step yields (PSY: g outgoing solids. 100 g^−1^ of incoming dried material) for the defatting step, SWE at 160 and 180 °C (soluble extracts: E-160 and E-180; insoluble residues: R-160 and R-180), and blanching treatment of insoluble residues (BR-160 and BR-180).

**Figure 2 foods-12-03759-f002:**
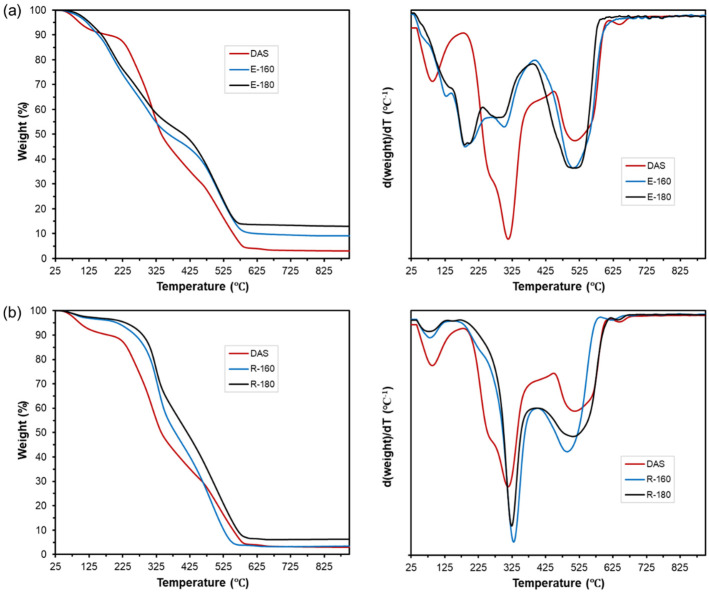
TGA and DTGA curves of the DAS, the active extracts (E-160 and E-180) (**a**), and the insoluble fractions (R-160 and R-180) (**b**) obtained from SWE at 160 and 180 °C.

**Figure 3 foods-12-03759-f003:**
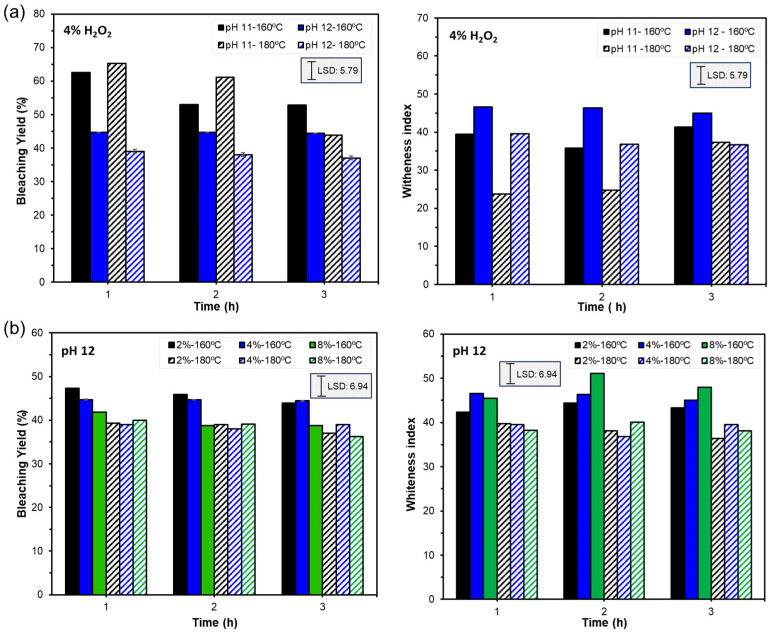
Values of bleaching yield and whiteness index reached in the different bleaching steps of the extraction residues obtained at 160 and 180 °C (R-160 and R-180 samples). (**a**) Treatments with 4% hydrogen peroxide, varying pH (11 and 12), and bleaching time (1, 2, and 3 h). (**b**) Treatments at pH 12, varying hydrogen peroxide concentration (2, 4, and 8%), and bleaching time (1, 2, and 3 h).

**Figure 4 foods-12-03759-f004:**
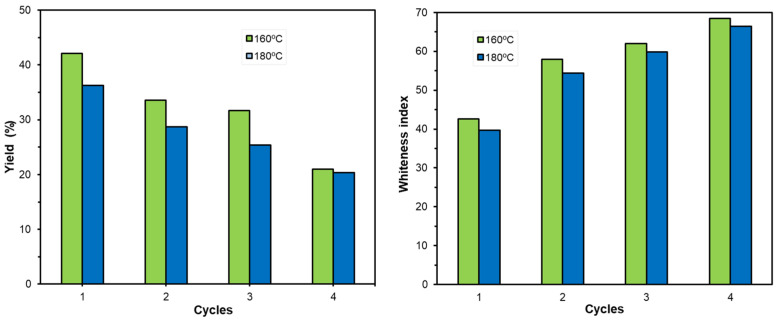
Values of bleaching yield and whiteness index reached in the different bleaching cycles of the extraction residues obtained at 160 and 180 °C (R-160 and R-180 samples).

**Figure 5 foods-12-03759-f005:**
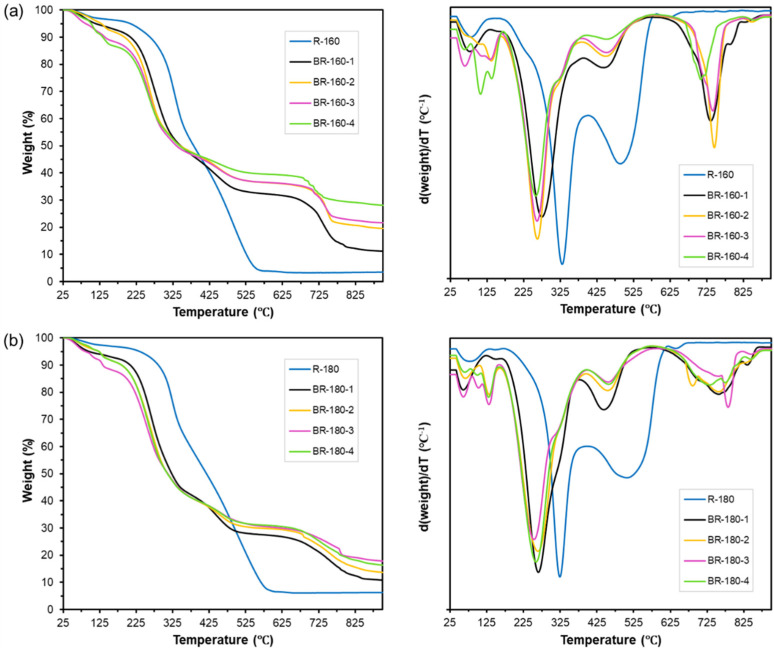
TGA and DTGA curves of the insoluble residues (R) and the bleached residues (BR) obtained from SWE at 160 (**a**) and 180 °C (**b**) submitted to different bleaching cycles (1–4).

**Table 1 foods-12-03759-t001:** Total phenolic (TPC) and ash contents, antioxidant activity (EC50), and minimal inhibitory concentration (MIC) of the DAS extracts (E-160 and E-180) (values ± standard deviation).

	E-160	E-180
TPC_1_ (mg GAE.g^−1^ dried extract)	100.9 ± 0.5 ^b^	161 ± 4 ^a^
TPC_2_ (mg GAE.g^−1^ DAS)	29.6 ± 0.15 ^b^	33.8 ± 0.9 ^a^
EC_50_ (mg extract.mg^−1^ DPPH)	1.490 ± 0.003 ^a^	1.063 ± 0.012 ^b^
EC_50_ (mg DAS.mg^−1^ DPPH)	5.13 ± 0.01 ^a^	5.06 ± 0.06 ^b^
MIC *L. innocua* (mg.mL^−1^)	90	34
MIC *E. coli* (mg.mL^−1^)	90	48
Ash (g.100 g^−1^ solid extract)	13.1 ± 0.1 ^b^	15.5 ± 0.7 ^a^

Different letters in the same row indicate significant differences between films by Fisher test (α = 0.05).

**Table 2 foods-12-03759-t002:** Chemical composition in terms of cellulose, hemicellulose, Klason lignin, and ash content of the defatted almond skin DAS, the insoluble residues (R-160 and R-180) obtained from SWE, and the bleached residues (BR-160 and BR-180) submitted to different bleaching cycles (1–4) (mean values ± standard deviation).

Sample	Cellulose (%wt.)	Hemicellulose (%wt.)	Klason Lignin (%wt.)	Ash (%wt.)
DAS	9.8 ± 0.3	11.6 ± 0.5	17.4 ± 0.2	5.1 ± 0.1
R-160	18.6 ± 0.9 ^b^	6.4 ± 0.3 ^a^	45.2 ± 0.5 ^b^	2.3 ± 0.1 ^a^
R-180	20.9 ± 0.2 ^a^	5.1 ± 0.1 ^b^	51.2 ± 0.3 ^a^	2.7 ± 0.6 ^a^
BR-160-1	35.0 ± 3.0 ^a,1^	13.4 ± 0.9 ^b,1^	22.0 ± 3.0 ^a,2^	15.0 ± 1.0 ^c,1^
BR-160-2	40.0 ± 4.0 ^a,2^	17.5 ± 1.9 ^a,1^	18.9 ± 1.9 ^b,2^	17.0 ± 4.0 ^bc,1^
BR-160-3	40.0 ± 4.0 ^a,2^	15.0 ± 3.0 ^ab,1^	15.0 ± 2.0 ^c,2^	18.0 ± 4.0 ^b,2^
BR-160-4	40.0 ± 3.0 ^a,2^	13.6 ± 0.7 ^ab,1^	13.6 ± 0.1 ^c,2^	25.0 ± 3.0 ^a,1^
BR-180-1	35.6 ± 1.8 ^b,1^	9.3 ± 1.2 ^b,2^	33.0 ± 2.0 ^a,1^	10.4 ± 0.1 ^c,2^
BR-180-2	52.0 ± 2.0 ^a,1^	13.5 ± 0.5 ^a,2^	22.5 ± 0.7 ^b,1^	17.0 ± 5.0 ^b,1^
BR-180-3	49.9 ± 0.8 ^a,1^	10.0 ± 2.0 ^b,1^	22.0 ± 4.0 ^bc,1^	26.3 ± 0.5 ^a,1^
BR-180-4	49.6 ± 0.6 ^a,1^	13.9 ± 1.2 ^a,1^	16.7 ± 0.7 ^c,1^	22.0 ± 7.0 ^ab,1^

Different subscript letters indicate significant differences between samples of the same group (insoluble residues R; bleached residues BR-160 or BR-180). Different numbers indicated significant differences between BR-160 and BR-180 samples at the same bleaching cycle (Fisher test, *p* < 0.05).

## Data Availability

The data used to support the findings of this study can be made available by the corresponding author upon request.

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
