# Peer review of "Subcritical Water Extraction for Valorisation of Almond Skin from Almond Industrial Processing"

_foods, 2023, doi:10.3390/foods12203759_

Round 1
Reviewer 1 Report
Comments for manuscript foods-2662024
The manuscript entitled “Subcritical water extraction for valorisation of almond skin from almond industrial processing” addresses an important issue. Food waste valorization is a great concern and directly related to Sustainable Development Goals of United Nations. Although the topic is of interest to the readers of the journal, the paper needs to be revised. Some recommendations are as the following:
- One of the major issues related to this manuscript is that the authors only employed non-specific spectrophotometric method for the determination of phenolic content. It is very well known from the literature that Folin-Ciocalteu assay is not specific to phenolic compounds. Reducing agents such as ascorbic acid, citric acid, simple sugars, or certain amino acids can interfere with the analysis and thus overestimate the content of phenolic compounds. Therefore, before drawing conclusions, chromatographic analysis of the phenolic compounds needs to be performed. This issue should be discussed in the manuscript.
- Other major issue is the use of a single antioxidant capacity measurement assay (DPPH). First of all, DPPH assay is criticized because of the involvement of biologically irrelevant free radicals and of the steric accessibility problems of the assay radicals by polymeric phenols. Secondly, this assay is only applicable to hydrophobic compounds. In addition, the measurement of antioxidant activity, in the case of multifunctional or complex multiphase systems cannot be evaluated satisfactorily by a single method. Even the methods based on the same principle can show several important differences in their response to antioxidants. Therefore, application of several different assays is highly recommended to evaluate antioxidant activities to obtain the full picture. I suggest authors to discuss this issue in the manuscript.
- In addition to the above, the manuscript needs to be revised in terms of language. Please ask a mother tongue to make complete check of the manuscript.
The manuscript needs to be revised in terms of language. Please ask a mother tongue to make complete check of the manuscript.
Author Response
Dear reviewers, thank you for your very accurate and useful comments to improve our manuscript. Please find below our answer to your comments.
The manuscript entitled “Subcritical water extraction for valorisation of almond skin from almond industrial processing” addresses an important issue. Food waste valorization is a great concern and directly related to Sustainable Development Goals of United Nations. Although the topic is of interest to the readers of the journal, the paper needs to be revised. Some recommendations are as the following:
- One of the major issues related to this manuscript is that the authors only employed non-specific spectrophotometric method for the determination of phenolic content. It is very well known from the literature that Folin-Ciocalteu assay is not specific to phenolic compounds. Reducing agents such as ascorbic acid, citric acid, simple sugars, or certain amino acids can interfere with the analysis and thus overestimate the content of phenolic compounds. Therefore, before drawing conclusions, chromatographic analysis of the phenolic compounds needs to be performed. This issue should be discussed in the manuscript.
The lack of specificity of the Folin-Cioclateu method was previously mentioned in the manuscript, but now it has been better described. A comment about the Folin-Ciocalteu method and its limitations and possible interferences has been added to the text. Nevertheless, this method can be useful to indicate the overall redox power of the extracts obtained under different extraction conditions, being related to their antioxidant power (comments added in lines 413-418). Chromatographic analyses of the almond skin phenolics have been extensively studied in the bibliography, as discussed in the Introduction section. It would be interesting to analyse the phenolics present in the extracts obtained in further studies, but it was not carried out at the moment. We only focused on the extraction process and the potential antioxidant properties of the extracts obtained using subcritical water extraction.
- Other major issue is the use of a single antioxidant capacity measurement assay (DPPH). First of all, DPPH assay is criticized because of the involvement of biologically irrelevant free radicals and of the steric accessibility problems of the assay radicals by polymeric phenols. Secondly, this assay is only applicable to hydrophobic compounds. In addition, the measurement of antioxidant activity, in the case of multifunctional or complex multiphase systems cannot be evaluated satisfactorily by a single method. Even the methods based on the same principle can show several important differences in their response to antioxidants. Therefore, application of several different assays is highly recommended to evaluate antioxidant activities to obtain the full picture. I suggest authors to discuss this issue in the manuscript.
Two methods were used to evaluate antioxidant capacity of the extracts (Folin-Ciocalteu and DPPH radical scavenging capacity) and coherent values were obtained from both methods. Other extensively reported assays with redox systems, such as ABTS radical scavenging capacity or FRAP method could be carried out. Nevertheless, for their use as active components of food packaging materials, their effectiveness at controlling food oxidation reactions should be proved when incorporated into these materials, to ensure the antioxidant power of the extracts at preventing oxidation of food components such as unsaturated lipids. This has been commented in the text (Lines 431-436).
- In addition to the above, the manuscript needs to be revised in terms of language. Please ask a mother tongue to make a complete check of the manuscript.
The manuscript was spell-checked and revised by a native English teacher.
Reviewer 2 Report
The paper entitled “Subcritical water extraction for valorisation of almond skin from 2 almond industrial processing” presents a possibility of using almond skin as source of valuable bioactive compounds.
Even though there are numerous studies conducted on the extraction of polyphenols from almonds by-products, the subject idea of the current exposed work brings information about the employment of water in its subcritical for the recovery of polyphenols and cellulose fibers.
The aims of the study are clearly expressed.
The experimental program is described in such manner that, with slight amendments, it can be easily applied.
The obtained results are concise, clearly presented and discussed.
Conclusions are drawn according to the obtained data.
The authors will find bellow some corrections and adjustments that should be addressed.
- The inclusion of the abbreviations meaning at their first appearance in the text (e.g. at lines 151 and 155 there is an abbreviation “DAP” for which there is no explanation included (or, perhaps it should be “DAS” instead of “DAP”).
- A more detailed explanation of why choosing the Escherichia coli and Listeria innocua strains apart of their apart from the fact that the first is gram negative and the second gram positive should be added.
- There are mentions about the fact that elevated temperatures may be responsible for the formation of HMF. For the particular case of the present work, what are the HMF levels?
- The “Materials and Methods” section provides data about the quantification of sugar content by HPLC. The “Results and discussion” section includes some brief information about the existence of sugars in DAS, R-160 and R-180 samples (see paragraph from lines 510 to 516) but there are no values given. Moreover, it is not very clear if the sugar quantification was made by NREL method, by HPLC or by both methods.
- The “Introduction” and “Conclusion” sections cite reinforced packaging materials as possible application of the obtained almond skin extracts. Essays were made in this direction or for the use of the extracts in other paths of interest? With what results?
Minor editing of English language required
Author Response
Dear reviewers, thank you for your very accurate and useful comments to improve our manuscript. Please find below our answer to your comments.
The paper entitled “Subcritical water extraction for valorisation of almond skin from 2 almond industrial processing” presents a possibility of using almond skin as source of valuable bioactive compounds.
Even though there are numerous studies conducted on the extraction of polyphenols from almonds by-products, the subject idea of the current exposed work brings information about the employment of water in its subcritical for the recovery of polyphenols and cellulose fibers.
The aims of the study are clearly expressed.
The experimental program is described in such manner that, with slight amendments, it can be easily applied.
The obtained results are concise, clearly presented and discussed.
Conclusions are drawn according to the obtained data.
The authors will find bellow some corrections and adjustments that should be addressed.
- The inclusion of the abbreviations meaning at their first appearance in the text (e.g. at lines 151 and 155 there is an abbreviation “DAP” for which there is no explanation included (or, perhaps it should be “DAS” instead of “DAP”).
The abbreviation DAP was replaced by DAS.
- A more detailed explanation of why choosing the Escherichia coli and Listeria innocua strains apart of their apart from the fact that the first is gram negative and the second gram positive should be added.
A more detailed explanation was added in the text (lines 438-442).
- There are mentions about the fact that elevated temperatures may be responsible for the formation of HMF. For the particular case of the present work, what are the HMF levels?
HMF levels were not quantified in this study, but these levels, as well as the adequate toxicological study should be carried out in further studies to ensure safety of the obtained extracts for practical uses, specifically as active ingredients for food packaging development, as commented on the final conclusion (Lines 627-639).
-The “Materials and Methods” section provides data about the quantification of sugar content by HPLC. The “Results and discussion” section includes some brief information about the existence of sugars in DAS, R-160 and R-180 samples (see paragraph from lines 510 to 516) but there are no values given. Moreover, it is not very clear if the sugar quantification was made by NREL method, by HPLC or by both methods.
The quantification of sugars was only carried out to determine structural carbohydrates (cellulose and hemicellulose) in DAS sample, the extraction residues (R-160 and R-180) and bleached residues applying 4 successive bleaching cycles, using the NREL. This method is based on two stages of acid hydrolysis of the lignocellulosic material and the quantification of the obtained simple sugars (from structural carbohydrates) by HPLC, as fully described in lines 188-208. In the text, we discuss this quantification in terms of structural carbohydrates, assuming the cellulose content as the glucose content and the hemicellulose content as the sum of arabinose and xylose contents, as reported in NREL method.
- The “Introduction” and “Conclusion” sections cite reinforced packaging materials as possible application of the obtained almond skin extracts. Essays were made in this direction or for the use of the extracts in other paths of interest? With what results?
This study is a first step towards the application of these extracts in different polymeric packaging materials to obtain active food packaging. Due to the high antioxidant and antimicrobial potential of the extracts, we hope that these materials will meet the requirements of active packaging and may extend the shelf life of packaged foods. So far, these studies are being carried out in the laboratory.
Reviewer 3 Report
The Authors report a study on the extraction of different chemicals from almond's skin through subcritical water extraction method.
The manuscript is well presented, written in a good English, and contains many interesting data. Several analytical techniques have been used and described.
Nevertheless, some minor issues are present and the manuscript could be improved.
"residue.100 g-1" This is maybe residue / 100 g or residue for 100 g?
Paragraph 3.1 During the discussion, should be useful for the reader if the Authors recall in the text the technique used. For example, in line 351 “the ash content in the DAS…… obtained how?
Can you add the anova table and some data related with the reliability of the model (as R-squared or other indicators)?
Lines 318-327: can you provide some experimental data in order to assess why in your case the mass balance is inconsistent? (gases formation or degradation)?
Line 331 “and/or”. The Authors should decide and prove if the signal is related to the loss of adsorbed water of to the degradation of sugars. Or if both the processes take place.
Author Response
Dear reviewers, thank you for your very accurate and useful comments to improve our manuscript. Please find below our answer to your comments.
The Authors report a study on the extraction of different chemicals from almond's skin through subcritical water extraction method.
The manuscript is well presented, written in a good English, and contains many interesting data. Several analytical techniques have been used and described.
Nevertheless, some minor issues are present and the manuscript could be improved.
"residue.100 g-1" This is maybe residue / 100 g or residue for 100 g?
This means the amount of insoluble fraction obtained per 100 g of deffated almond skin used in the extraction. The word "residue" has been replaced by "insoluble fraction" (lines 303-304).
Paragraph 3.1 During the discussion, should be useful for the reader if the Authors recall in the text the technique used. For example, in line 351 “the ash content in the DAS…… obtained how? It has been done (lines 352-353).
Can you add the anova table and some data related with the reliability of the model (as R-squared or other indicators)?
As concerns statistical indicators of MANOVA, we have included the F-values of each factor on the response variables (BY and WI) in the text (lines 475-508), as well as the LSD values from Fisher's test, for each response variable analysed, in Figure 3.
Lines 318-327: can you provide some experimental data in order to assess why in your case the mass balance is inconsistent? (gases formation or degradation)?
This was discussed in lines 304-318.
“It is also remarkable that the sum of both yields at a given temperature does not close the mass balance, in which about 15 % and 24 % of dry matter was not recovered at 160 ℃ and 180 ℃, respectively. On the other hand, the pressure in the extraction reactor exceeded the water vapour pressure at the set point temperature by 2 and 5 bars, for 160 and 180 ℃, respectively. This result suggests the formation of gases associated with a certain degree of mineralisation of the organic matter present in the reactor at the process conditions, forming CO2. In fact, SWE has been applied with high efficiency in the mineralisation of contaminant organic compounds, using hydrogen peroxide at different concentrations to effectively remove many hazardous compounds in wastewater [43,44]. Although in the extraction carried out, no oxidising agent was added, and it took place at lower temperatures, the demonstrated prooxidant action of many phenolic compounds [45], depending on the medium conditions, could induce partial mineralisation of DAS samples under the subcritical water conditions applied in the study; the higher the extraction temperature, the greater the substrate degradation.”
Line 331 “and/or”. The Authors should decide and prove if the signal is related to the loss of adsorbed water of to the degradation of sugars. Or if both the processes take place.
Both process occurred this temperature range. This has been changed in the text (lines 332 and 337).
Round 2
Reviewer 1 Report
The authors addressed all the points that are raised by the reviewer. The manuscript is now improved and therefore may be accepted for publication.